# Carrier-Mediated Drug Uptake in Fungal Pathogens

**DOI:** 10.3390/genes11111324

**Published:** 2020-11-09

**Authors:** Mónica Galocha, Inês Vieira Costa, Miguel Cacho Teixeira

**Affiliations:** 1Department of Bioengineering, Instituto Superior Técnico, Universidade de Lisboa, 1049-001 Lisboa, Portugal; monicagalocha@tecnico.ulisboa.pt (M.G.); costa.ines4@gmail.com (I.V.C.); 2Biological Sciences Research Group, iBB-Institute for Bioengineering and Biosciences, Instituto Superior Técnico, Universidade de Lisboa, 1049-001 Lisboa, Portugal

**Keywords:** drug uptake, nutrient permeases, phylogenetic analysis, fungal pathogens, drug resistance

## Abstract

*Candida*, *Aspergillus*, and *Cryptococcus* species are the most frequent cause of severe human fungal infections. Clinically relevant antifungal drugs are scarce, and their effectiveness are hampered by the ability of fungal cells to develop drug resistance mechanisms. Drug effectiveness and drug resistance in human pathogens is very often affected by their “transportome”. Many studies have covered a panoply of drug resistance mechanisms that depend on drug efflux pumps belonging to the ATP-Binding Cassette and Major Facilitator Superfamily. However, the study of drug uptake mechanisms has been, to some extent, overlooked in pathogenic fungi. This review focuses on discussing current knowledge on drug uptake systems in fungal pathogens, highlighting the need for further studies on this topic of great importance. The following subjects are covered: (i) drugs imported by known transporter(s) in pathogenic fungi; and (ii) drugs imported by known transporter(s) in the model yeast *Saccharomyces cerevisiae* or in human parasites, aimed at the identification of their homologs in pathogenic fungi. Besides its contribution to increase the understanding of drug-pathogen interactions, the practical implications of identifying drug importers in human pathogens are discussed, particularly focusing on drug development strategies.

## 1. Introduction

Drug transporters are considered key determinants of drug activity by affecting absorption, distribution, and excretion of drugs in patients under treatment [1]. Therefore, in an effort to improve drug efficacy, transporter-mediated drug delivery in human cells have been studied in past decades [2,3]. Generally, solute carrier (SLC) transporters, including members of the major facilitator superfamily (MFS), mediate solute influx, while ATP-binding cassette (ABC) transporters mediate solute efflux [2]. Although the transport mechanisms for the majority of the drugs remain to be clarified, it has been demonstrated that differences in the expression level of drug transporters in different tissues contribute to variability in pharmacokinetic parameters and drug effectiveness [4]. The increasing awareness that drug uptake is, more frequently than expected, carrier-mediated, raises the hypothesis that many drug-carrier pairs remain to be identified [5].

Drug resistance in concerning human pathogens, such as species of *Candida*, *Aspergillus* and *Cryptococcus*, which are the most frequent cause of severe fungal infections [6], and of *Trypanosoma* and *Leishmania*, which are among the most common causes of protozoa-associated human diseases [7], is very often achieved by changes in their “transportome”. The increased extrusion of drugs through the upregulation of drug efflux pumps is considered to be a key mechanism of drug resistance in some of these human pathogens and several families of efflux systems capable of multiple drug efflux have been described [8,9,10,11]. Similarly, decreased drug uptake caused by alterations on drug-carriers [12,13] is a common and quite well characterized drug resistance mechanism in human parasites [7]. However, the entry of drugs into fungal cells has conventionally been thought to occur primarily via passive diffusion through the cell membrane depending on drug lipophilicity [14,15]. Nevertheless, increasing evidence hints to the possibility that, at least some antifungal drugs of both hydrophilic and hydrophobic nature, are imported into cells through nutrient permeases [5]. On the one hand, some studies have shown that specific drugs are transported into cells, although the identification of the responsible carrier(s) failed or was not inspected [16,17]. On the other hand, there is evidence that impaired activity of specific carriers leads to a decreased sensitivity of cells when exposed to specific xenobiotics, strongly suggesting that such carriers are responsible for drug uptake [5,18,19].

Drugs frequently enter cells as opportunistic substrates of transporters, whose physiological role is the import of biologically relevant solutes. In fact, a large number of the identified drug carriers are nutrient permeases, which are hijacked by drugs to cross the cytoplasmic membrane [5,20]. A well-characterized example is the case of the pyrimidine analog 5-flucytosine (5-FC), which was the first drug used to treat fungal infections in the 1960s. 5-FC enters fungal cells through a cytosine permease, normally present in the cell membrane, and once inside fungal cells it is processed through the pyrimidine salvage pathway originating a compound that interferes with both RNA and DNA synthesis [21,22,23]. Some nutrient permeases have a quite broad substrate specificity, such as the promiscuous general amino acid permeases, which are capable of transporting all types of amino acids, ranging from hydrophilic to hydrophobic, small to aromatic, positively to negatively charged [24]. Several amino acids contain a plethora of potentially reactive ligands; therefore, their transporters offer excellent potential portals of entry for new drugs. This points to the idea that it might be of particular interest to identify broad-spectrum nutrient permeases, which can be more easily hijacked by xenobiotic compounds.

Despite few very good one-off efforts [16,17], the study of drug uptake mechanisms has been, to some extent, overlooked in pathogenic fungi. Given that altered drug uptake is an important mechanism of drug effectiveness in human parasites [7], and considering the increasing number of drug resistant clinical isolates of pathogenic fungi, for which the underlying drug resistance mechanisms are unknown [25,26], extending the knowledge on this process to pathogenic fungi is of great importance.

This review explores not only the scarce current knowledge regarding drug uptake in pathogenic fungi, but also the clues that may be retrieved from the model yeast *Saccharomyces cerevisiae* and from human protozoa pathogens of the *Leishmania* and *Trypanosoma* genera. The following subjects are covered: (i) drugs imported by known transporter(s) in pathogenic fungi; and (ii) drugs imported by known transporter(s) in the model yeast *Saccharomyces cerevisiae* or in human parasites, aiming the identification of their homologs in pathogenic fungi. Besides contributing to the understanding of drug-pathogen interactions, the identification of drug importers in human pathogens has two important applications: (1) the identification of potential drug resistance mechanisms, involving the acquisition of mutations that lead to decreased drug uptake; and (2) the use/design of drugs that can hijack nutrient permeases, thus, increasing the likelihood of a drug to accumulate intracellularly. If the selected permeases are essential membrane proteins, drug resistance through the evolution towards decreased drug uptake would be prevented, as it would decrease essential nutrient uptake, compromising cell viability.

## 2. Drugs Imported by Known Transporter(s) in Pathogenic Fungi

It is estimated that there are a hundred times more diverse compounds than membrane transporters in nature and, accordingly, considerable transporter promiscuity is both expected and mandatory [5]. However, the number of studies on drug uptake in fungal pathogens is scarce. The few known cases of characterized drug uptake proteins are described below.

5-FC active uptake is the best studied case of antifungal drug import in fungal pathogens. 5-Flucytosine is a pro-drug that uses the salvage pathway of pyrimidine to exert its toxic activity, entering the fungal cell through, at least, the purine/cytosine permease Fcy2 [5,22,27,28]. This has been observed in *Candida albicans* [29], which also has two other purine-cytosine permeases, Fcy21 and Fcy22, highly homologous to Fcy2 [22]. In *Candida albicans* clinical isolates, amino acid substitutions in Fcy21 and Fcy22 were found in 5-FC resistant strains, which suggests the ability of Fcy21 and Fcy22 to transport 5-FC as well [22,30]. Fcy2 also mediates 5-FC uptake in *Candida glabrata* and *Candida lusitaniae* [21,31,32]. In *Candida lusitaniae,* there is also evidence that, the uracil permease Fur4 could transport 5-fluorouracil, the deaminated product of 5-FC [27]. It is known that in *Cryptococcus* species, Fcy2 is also the mediator for 5-FC import, namely in *Cryptococcus gattii* [33] and *Cryptococcus neoformans* [34]. Finally, in *Aspergillus nidulans* and *Aspergillus fumigatus*, 5-FC uptake is thought to be carried out by FcyB, encoding a purine-cytosine permease orthologue to *Candida albicans* Fcy2, member of the high-affinity nucleobase cation symporter family (NCS1) [23,35].

Visceral leishmaniosis (VL)-2397 (ASp2397), a novel antifungal drug with a mechanism of action that is not yet fully understood, is also known to enter fungal cells through active import. Its structure is similar to that of ferrichrome, a siderophore with high specificity for iron. VL-2397 was shown to have a strong antifungal activity against *Aspergillus* species, and also against other fungal species including *Candida glabrata* and *Cryptococcus neoformans* [36]. This drug was also demonstrated to be effective against azole resistant strains of *Aspergillus fumigatus* and *Aspergillus terreus*, and to have a faster, and more potent fungicidal effect in *Aspergillus fumigatus* germinated conidia and a stronger inhibition of hyphal elongation, than other antifungal agents like amphotericin B or azole drugs, such as posaconazole and voriconazole [36]. In *Aspergillus fumigatus*, VL-2397 uptake was shown to be catalyzed by an importer for ferrichrome-type siderophores, Sit1, whose inactivation results in resistance to VL-2397 [37]. In contrast to the referred fungal species, VL-2397 is not effective against *Saccharomyces cerevisiae*. Yet, expression of Sit1 from *Aspergillus fumigatus*, or of its *Candida glabrata* homolog, renders susceptibility to VL-2397 in this model yeast, which suggests that the intrinsic resistance of *Saccharomyces cerevisiae* is based on lack of drug uptake.

A third example of an antifungal drug known to be actively imported into yeast cells is that of bis [1,6-a:5′,6′-g] quinolizinium-8-methyl-acetate (BQM). BQM interacts with mitochondria, inducing reactive oxygen species (ROS) production to exert its antifungal activity. Surprisingly, BQM import was found to depend on the expression of the Mdr1 transporter, a widely known multidrug efflux pump whose overexpression in *Candida albicans* drug-resistant clinical isolates confers fluconazole resistance [38]. It was suggested that Mdr1 might mediate a bidirectional transport depending on the substrate, and to mediate the direct influx of BQM [38].

## 3. Drugs Imported by Known Transporter(s) in the Model Yeast *Saccharomyces cerevisiae* or in Human Parasites—Their Homologs in Pathogenic Fungi

There is evidence of carrier-mediated import for a wide range of molecules in the model yeast *Saccharomyces cerevisiae*, and also in human parasites, including molecules that are clinically used against fungal infections or that were reported to be effective against the most concerning fungal species. Such cases are reviewed in this section. Moreover, it is reasonable to hypothesize that homologs of these transporters in pathogenic fungi may share the same function. Thus, a phylogenetic analysis of known *Saccharomyces cerevisiae* and some human parasites drug transporters and their closest homologs in pathogenic fungi was pursued. This analysis might provide a ground start for the study of drug uptake in pathogenic fungi and, consequently, explore those mechanisms as viable targets for drug development or optimization.

To search for homolog proteins of *Saccharomyces cerevisiae* in the pathogenic *Candida albicans* and *Candida glabrata*, the YEAst Search for Transcription Regulators And Consensus Tracking (YEASTRACT) + bioinformatic tool was used [39]. To determine the homolog proteins, this handy bioinformatic tool uses the proteome of each species as the input for a protein Basic Local Alignment Search Tool (BLASTp) [40], performed in a reciprocal way between all the species of the database. The BLASTp hit with the highest score for each protein sequence is considered as the best hit. Additionally, a tolerance of 10% is applied, meaning that alignments with a score almost identical to the best-hit are not lost. Following this bidirectional BLASTp analysis, the homolog proteins of human parasites in pathogenic fungi *Candida albicans*, *Candida glabrata*, *Cryptococcus neoformans*, and *Aspergillus fumigatus* were also determined as follows: using the protein sequence of known parasite drug importers as the input for a BLASTp, best hits on the different fungal species were selected; then, using those best hits protein sequence as the input for a BLASTp, the best hit on the respective parasite proteome was determined; if it matches, those fungal hits were considered as homologs of the parasite transporter protein. When protein name/Open Reading Frame (ORF) was not available, GenBank [41] number was used as identifier.

The evolutionary history of the gathered protein families was inferred by using the Maximum Likelihood method and Jones-Taylor-Thornton (JTT) matrix-based model [42]. The trees with the highest log likelihood, for each case, are shown below. Initial tree(s) for the heuristic search were obtained automatically by applying neighbor-joining and Bio Neighbor-Joining (BioNJ) algorithms to a matrix of pairwise distances estimated using the JTT model, and then selecting the topology with superior log likelihood value. The proportion of sites where at least 1 unambiguous base is present in at least 1 sequence for each descendent clade is shown next to each internal node in the tree. Evolutionary analyses were conducted in MEGA X (https://pubmed.ncbi.nlm.nih.gov/29722887/) [43].

### 3.1. Clues from the Model Yeast Saccharomyces cerevisiae

The clade Ascomycota is the largest phylum of fungi [44] and includes, besides the model yeast *Saccharomyces cerevisiae* (Saccharomycotina lineage), most animal parasitic fungi, including *Candida* (Saccharomycotina lineage) and *Aspergillus* (Pezizomycotina lineage) species [45]. Most pathogenic *Candida* species, including *Candida albicans*, *C. parapsilosis*, and *C. tropicalis*, belong to the so-called CTG clade. However, among the non-*Candida albicans Candida* species, *Candida glabrata* is more closely related to *Saccharomyces *cerevisiae**, both belonging to the Whole Genome Duplication (WGD) clade of the Saccharomycetaceae [46]. Although *Saccharomyces cerevisiae* and *Candida glabrata* are two closely related yeast species at an evolutionary scale, their different habitats and lifestyles correlate with specific gene differences and with more extensive gene losses having occurred in *Candida glabrata* [47].

Apart from Ascomycota, the Basidiomycota is the second most species-rich phylum of fungi and harbors the next most important agents of human disease, *Cryptococcus* species that are opportunistic pathogens with particular impact in AIDS patients (i.e., *Cryptococcus neoformans*) [48]. Despite their phylogenetic divergence, the comparison of the genomes of *Saccharomyces cerevisiae* and *Cryptococcus neoformans* demonstrates that they share at least 65% of the genetic information [49]. Obviously, some genetic traits, such as those coding for capsule synthesis and other possible virulence factors, are specific of *Cryptococcus neoformans*. This high level of conservation is remarkable for two species that diverged in the evolutionary tree about 1000 million years ago [50].

*Saccharomyces cerevisiae* is one of the most widely used eukaryotic model organisms. It has largely contributed to the understanding of a number of cellular mechanisms, such as aging, regulation of gene expression, cell cycle, and neurodegenerative disorders, among others. In fact, up to 30% of the genes associated with known human diseases have counterparts in the yeast genome [51]. The advantages of *Saccharomyces cerevisiae* include highly developed molecular biology tools for genetic modification, including a genome-wide ‘knockout’/‘knockdown’ library of all genes, plus very well characterized genome, proteome, and metabolome information [52]. Bioinformatic analysis predicts that the *Saccharomyces cerevisiae* genome contains 318 transporter proteins [53], this “transportome” having been extensively studied. Altogether, this provides an ideal background for the development and validation of uptake transporter screenings. Indeed, Lanthaler et al. [5] performed a high throughput experiment to screen yeast strains with single deletions of each of the 111 genes encoding plasma membrane transporters. They were able to suggest an association between 18 cytotoxic compounds and at least one known yeast transporter, based on the observation that the deletion of the encoding gene leads to altered xenobiotic resistance. Among those 18 compounds, 12 have been previously shown efficacy against some of the fungal pathogens, including the widely used antifungal azole drugs fluconazole, ketoconazole, and clotrimazole. The role of these 12 transporters is detailed bellow, aiming to predict homologous fungal pathogen uptake systems.

As the first example, the deletion of the myo-inositol transporter *Sc*Itr1 rendered increased yeast cells resistance to some of the most widely used azoles, suggesting that this transporter might be involved in the uptake of ketoconazole, clotrimazole, and fluconazole—molecules that are quite divergent from *Sc*Itr1 natural substrate [5]. Interestingly, it had been reported that fluconazole enters fungal cells by carrier facilitated import in *Saccharomyces cerevisiae*, *Candida albicans*, *Candida krusei*, *Cryptococcus neoformans,* and *Aspergillus fumigatus* [16,17]; however, the responsible transporter(s) failed to be identified. Remarkably, all of the fungal pathogens considered in this review express homologs of the *Sc*Itr1 transporter. As expected, considering the phylogenetic distance between the fungal species addressed, the closest homolog to *Sc*Itr1 is encoded by an uncharacterized ORF from *Candida glabrata* (*CAGL0I07447g*) (Figure 1), which is actually more closely related to the *Sc*Itr1 paralog, *Sc*Itr2. The following is an MFS inositol transporter from *Candida albicans*, *Ca*Itr1 (Figure 1). Interestingly, *CaITR1* expression was found to be repressed in the presence of fluconazole [54], which might indicate a link between this transporter and azole uptake, but this has not yet been demonstrated. The closest *Sc*Itr1 homologs from *Aspergillus fumigatus* and *Cryptococcus neoformans* are not characterized, but deserve evaluation in this context.

The second example of a yeast transporter involved in drug uptake is the high-affinity nicotinamide riboside transporter *Sc*Nrt1. *Sc*Nrt1 was shown to import diphenyleneiodonium chloride (DPI), an inhibitor of the superoxide producing enzyme NADPH oxidase that impairs fungal spore germination [55], and methotrexate, an anti-tumor drug of the folate class that showed a synergistic effect when used together with azoles against *Candida albicans*-associated invasive candidiasis [56]. Unlike what was observed for *Sc*Itr1, homolog proteins for *Sc*Ntr1 transporter were only found in the closely related pathogenic yeast *Candida glabrata*. The closest *Sc*Ntr1 homologs are uncharacterized proteins, predicted to be a thiamine transporter, *Cg*Thi7 (Figure 2), a high-affinity transporter of NAD+ precursors, *Cg*Tnr2, and a putative nicotinamide transporter, *Cg*Tnr1 (Figure 2). To the best of our knowledge, none of these three proteins have been associated with drug transport so far.

The third example is *Sc*Fen2, a pantothenate transporter whose deletion rendered a resistance phenotype to 5-fluorouracil (5-FU) in *Saccharomyces cerevisiae* [5]. The characteristic carboxyl group of Fen2 substrates is absent in this molecule, which suggests that Fen2 might transport a wider range of substrates than initially foreseen. The closest homolog of *Sc*Fen2 is an uncharacterized predicted pantothenate transporter from *Candida glabrata*, followed by a predicted MFS transporter member of the anion/cation symporter (ACS) family from *Candida albicans* (Figure 3). Similarly, the closest homolog proteins from *Aspergillus fumigatus* and *Cryptococcus neoformans* are predicted pantothenate transporters, none of which has been characterized yet.

The fourth example of a predicted *Saccharomyces cerevisiae* drug uptake transporter is the low affinity amino acid permease *Sc*Agp1. ScAgp1 is known to have a broad substrate range and was shown to import mitoxantrone, an antineoplastic agent that acts by inhibiting Type II topoisomerases and was shown to have antifungal activity against *Candida glabrata* and *Cryptococcus neoformans* [57], and to inhibit morphogenesis in *Candida albicans* [58]. This transporter has a paralog that arose from the whole-genome duplication, *Sc*Gnp1, which was included in the phylogenetic analysis. The closest homolog proteins in *Candida glabrata* are two uncharacterized proteins, closer to *Sc*Gnp1 than to *Sc*Agp1. Surprisingly, the expression of the closest homolog protein from *Candida albicans*, *Ca*Gnp1 (Figure 4), was found to be induced under the presence of fluconazole or caspofungin [54]. This might suggest that this permease is able to affect drug resistance, eventually through the import and/or export of distinct compounds. No *Sc*Agp1 homolog proteins were found in *Cryptococcus neoformans* or *Aspergillus fumigatus*.

The fifth example is the uridine permease *Sc*Fui1, suggested to be an importer of benzbromarone, which is a uricosuric agent used in the treatment of gout, shown to be effective at inhibiting *Candida albicans* biofilm formation and growth. Since benzbromarone is a strong inhibitor of cytochrome P450 isoform CYP2C9 in mammalian liver, it is speculated that it may inhibit fungal CYP51, similar to azoles, which is essential for the synthesis of ergosterol [59]. This drug is structurally very similar to the uridine substrate of *Sc*Fui1, which might explain its role in drug uptake. The phylogenetic analysis demonstrated that there are no *Sc*Fui1 homolog proteins in *Aspergillus fumigatus*, and the closest homolog proteins in the remaining fungal species have not yet been characterized or associated to drug transport (Figure 5).

The above addressed transporters belong to the fungal NCS1 subfamily, which includes various *Saccharomyces cerevisiae* transporters: nicotinamide riboside transporter 1 (Nrt1, also called Thi7), Dal4 (allantoin permease), Fui1 (uridine permease), Fur4 (uracil permease), and Thi7 (thiamine transporter). NCS1 transporters are essential components of salvage pathways for nucleobases and related metabolites. NCS1s belong to a superfamily which also contains the solute carrier 5 family sodium/glucose transporters, and solute carrier 6 family neurotransmitter transporters [60]. Given these indications, it would be interesting to further explore the potential role of this transporter family in drug resistance or, eventually, as new drug targets.

Another example of a proposed drug mediated uptake process is connected to cisplatin, an anticancer drug that was shown to be able to inhibit *Candida* and *Aspergillus* filamentation/conidiation. Pretreatment of *Candida albicans* with amphotericin B (and miconazole) increases its sensitivity to cisplatin in vitro, suggesting a degree of synergistic antifungal activity of these agents [61]. Surprisingly, this drug was suggested to be imported by several transporters in *Saccharomyces cerevisiae*, namely, Fcy2, which is known to import 5-FC, Hnm1, and Lem3 [5].

*Sc*Hnm1 is a choline/ethanolamine transporter proposed to be involved in cisplatin uptake in *Saccharomyces cerevisiae*. Clear homologs of this protein are expressed in *Candida albicans*, *Candida glabrata*, *Cryptococcus neoformans,* and *Aspergillus fumigatus*, most of whom are predicted choline transporters (Figure 6). Their possible role in drug uptake, suggested herein, remains to be established.

Lem3, a phospholipid transporter, was found to also catalyze the uptake of alkylphosphocholine drugs, including miltefosine, an antiparasitic drug used for leishmaniosis treatment [24,62], that enters into parasitic cells by carrier-mediated transport [63]. Lem3 was also shown to mediate the transport of tunicamycin, an antibiotic that inhibits protein *N*-glycosylation [5]. Interestingly, physiological concentrations of tunicamycin were shown to display significant inhibitory effects on *Candida albicans* biofilm development and maintenance, while not affecting overall cell growth or morphology [64]. Moreover, the combination of tunicamycin and amphotericin B was shown to reduce *Candida albicans* growth in vitro [65].

*Sc*Lem3 homologs were found in the two *Candida* species under study, but not in *Cryptococcus neoformans* or in *Aspergillus fumigatus* (Figure 7). The closest *Sc*Lem3 homolog in *Candida glabrata*, *Cg*Lem3, is a glycoprotein that is also involved in the membrane translocation of phospholipids and alkylphosphocholine drugs [66]. Interestingly, disruption of the *Sc*Lem3 homolog from *Candida albicans*, a putative membrane protein *Ca*Lem3 (Figure 7), was shown to increase resistance to miltefosine [66], reinforcing the idea that transporter function might be conserved across species.

*Sc*Qdr2, a plasma membrane MFS transporter with broad substrate specificity, has also been linked to the uptake of tunicamycin, given the observation that its deletion confers tunicamycin resistance [5]. This was a surprising observation, as *Sc*Qdr2 has been mostly characterized as a drug exporter, involved in the export of the antiarrhythmic drug quinidine [67], and in conferring resistance to numerous other compounds, including cisplatin and bleomycin [68], as well as polyamines [69]. Interestingly, however, *Sc*Qdr2 was further found to catalyze the uptake of K^+^ [70]. *Sc*Qdr2 homologs (Figure 8), were also found to confer drug resistance. *Cg*Qdr2, which is the closest *Sc*Qdr2 homolog, is associated to imidazole resistance and export [71], while affecting plasma membrane potential and promoting biofilm formation, likely in an indirect fashion [72]. On the other hand, *Ca*Qdr1, was found to play no role in antifungal drug resistance, while sharing with its *Candida glabrata* homolog an effect in biofilm formation and virulence [73]. *Ca*Qdr1 was found to be repressed under caspofungin treatment [54], which might suggest that it could play a role in caspofungin import; however, the deletion of *CaQDR1* was found to have no effect in caspofungin susceptibility [73]. The *Sc*Qdr2 closest homolog in *Aspergillus fumigatus* (sequence ID: XP KEY77771.1) (Figure 8), is a MFS protein downregulated in response to amphotericin B [74]. No *Sc*Qdr2 homolog could be found in *Cryptococcus neoformans*. Altogether, although it is unlikely that Qdr2 proteins play a clear role in drug uptake, it would be interesting to evaluate this hypothesis.

Hexose transporters (HXT) in *Saccharomyces cerevisiae* have also been linked to the uptake of drugs. Hxt9 and Hxt11 were shown to confer susceptibility to cycloheximide, sulfometuron methyl, and 4-NQO (4-nitroquinoline-*N*-oxide) [75], while Hxt1, Hxt5, Hxt8, and Hxt10 confer quinine susceptibility [76], in all cases suggesting a role in the uptake of the corresponding drugs. The hexose transporter family is a very large one, spread through all yeasts and molds. Nonetheless, considering the Hxt transporters that were so far related with drug import in *Saccharomyces cerevisiae*, clear homologs were only found in *Candida albicans* and *Candida glabrata* (Figure 9). All of these transporters may be interesting as potentially linked to drug import, a phenotype that has failed to be explored in pathogenic fungi.

### 3.2. Clues from Human Protozoa Pathogens

The armamentarium of antiprotozoal drugs is limited, and the effectiveness of these drugs is decreasing because of the widespread resistance development. In the majority of the cases, transporters were found to be crucial for sensitivity and resistance to these drugs [7]. Loss or alteration of uptake transporters appear as one of the strategies of acquired drug resistance in protozoa [77]. Although protozoa and fungal organisms are quite distant phylogenetically, comparison of specific cell surface proteins, for instance, reveals a number of common traits [78]. In fact, as referred above, an *Saccharomyces cerevisiae* phospholipid transporter, Lem3, was shown to be able to import miltefosine, a drug used for leishmaniosis treatment, into yeast cells. This emphasizes the idea of transporter promiscuity and of valuable phylogenetic comparison to find similar transporters in different human pathogens.

#### 3.2.1. Leishmania

*Leishmania* spp. parasites cause a systemic infection called visceral leishmaniosis (VL), which is almost always fatal if left untreated. For almost seven decades, pentavalent antimonials were the standard antileishmanial treatment worldwide. However, during the last 15 years, their clinical value was threatened due to the widespread emergence of resistance. In the last decade, the first orally administrated drug for the treatment of VL, miltefosine (hexadecylphosphocholine), became available [62]. This drug belongs to the alkylphosphocholine class and the exact mechanisms of action and resistance remain largely unknown. However, a decrease in drug accumulation has been reported for all miltefosine resistant *Leishmania* lines studied [79]. The internalization of miltefosine in the parasite cell occurs through the miltefosine membrane transporter LdMT and its β subunit LdRos3. Once inside the cell, the drug induces apoptosis-like cell death [80]. In fact, experimental mutations that inactivate LdMT or LdRos3 rendered the parasites remarkably less sensitive to miltefosine, and this resistance persisted in vivo; cross-resistance with other antileishmanials was not detected [63,80].

Interestingly, besides being primarily used as an antiparasitic drug, miltefosine has considerable in vitro antifungal potential against a wide range of pathogenic fungi, including *Cryptococcus* spp. [81], *Candida* spp. [82,83] and *Aspergillus* spp. [84]. More important, when tested under planktonic conditions, miltefosine displays potent in vitro activity against multiple fluconazole-susceptible and -resistant *Candida albicans* clinical isolates, including isolates overexpressing efflux pumps and/or with well-characterized *ERG11* mutations [82]. Moreover, this drug inhibits *Candida albicans* biofilm formation and displays activity against preformed biofilms. This is a major trait, since biofilm formation by pathogenic fungus is usually associated with increased resistance to antifungals and treatment failures. Although the mechanism of internalization and action of miltefosine in fungi is unknown, this drug represents a potential alternative to conventional therapy for fungal infections. Since it is known that the uptake of miltefosine in *Leishmania* spp. is achieved through the action of a membrane transporter, LdMT, and that its action is crucial for resistance development, it is worth to identify possible homolog transporters in concerning human fungal pathogens.

Almost all the closest LdMT homologs in the fungal species under analysis are associated with phospholipid transport across the membrane. The closest homolog of LdMT in pathogenic fungi seems to be the uncharacterized proteins OXC66518.1 and OWZ80564.1 from *Cryptococcus neoformans* (Figure 10). Additional homologs are also found in *Candida albicans*, *Candida glabrata*, and *Aspergillus fumigatus*. Their potential role in miltefosine uptake should be evaluated if this drug is ever to be used in antifungal therapy.

#### 3.2.2. Trypanosoma

The genus *Trypanosoma* contains about 20 species, but only *T. cruzi* and the two African trypanosome subspecies, *Trypanosoma brucei gambiense* and *T. b. rhodesiense*, cause disease in humans [85]. *T. brucei* is an extracellular parasite carried around in the bloodstream of infected hosts. It is transmitted by the tsetse fly and causes African sleeping sickness [78]. In trypanosomiasis-affected regions, current therapies are characterized by high toxicity and increasing drug resistance associated, at least in part, with loss-of-function mutations in the transporters involved in drug import. Several studies have surveyed the *T. brucei* genome to identify proteins that are involved in the transport of chemotherapeutic agents [86,87,88]. These have identified loss-of-function mutations in nutrient transporters as the underlying mechanisms of drug resistance.

Pentamidine and melarsoprol are part of the limited weapons available to treat Human African trypanosomiasis (HAT), and resistance have been found to be related with loss of function and mutations in the transporters responsible for their uptake [7]. Melarsoprol is an arsenical drug that interacts with thiol groups of several key proteins, depriving the parasite of its main sulfhydryl antioxidant and inhibiting trypanothione reductase, also depriving the parasite of the essential enzyme system that is responsible for keeping trypanothione reduced [89]. This drug enters the parasite via an adenosine transporter, TbAT1 [90], and an aquaglyceroporin, AQP2 [91]. Pentamidine, like melarsoprol, enters trypanosomes via AQP2 [91], but also via NT11.1/AT-A and NT12.1/AT-E, which are adenine transporters [92]. Pentamidine-melarsoprol cross-resistance (MPXP) is a major concern for HAT treatment, being caused by mutations in the adenosine transporter TbAT1 and in the aquaglyceroporin AQP2. AQP2 is also responsible for the entry of boric acid into trypanosome cells, along with the aquaglyceroporins AQP3 and AQP1 [93]. Boric acid is an effective treatment for yeast infection. For instance, it is a safe, alternative, economic option for women with recurrent and chronic symptoms of vaginitis when conventional treatment fails because of the involvement of non-*albicans Candida* spp. or azole-resistant strains [94].

BLAST did not found any similarity between the TbAT1 adenosine transporter or the NT11.1/AT-A and NT12.1/AT-E adenine transporters and proteins from the fungal pathogens addressed in this review. Regarding the aquaglyceroporin AQP2, BLASTp found significant hits only in *Aspergillus fumigatus* and *Candida glabrata* (Figure 11), although none has yet been associated with drug import. On the contrary, the *Candida glabrata* Fps1 (ORF *CAGL0C03267g*) and Fps2 (ORF *CAGL0E03894g*) aquaglyceroporins were identified as determinants of 5-FC resistance, decreasing drug accumulation in *Candida glabrata* cells [95]. Their role in caspofungin resistance was also reported, although not being associated with drug import/export [96].

Eflornithine is an irreversible inhibitor for ornithine decarboxylase, an essential enzyme for the first step of polyamines synthesis and the formation of the trypanothione [97]. The drug crosses the blood-brain barrier and its entry into trypanosomes is also transporter-mediated. The amino acid transporter TbAAT6 was identified as the responsible for eflornithine uptake and its inactivation is associated with drug resistance [92,98]. However, this mechanism of resistance to eflornithine still remains to be confirmed in resistant parasites from clinical isolates. In this case, it was also not possible to identify homolog proteins in fungal species.

## 4. Conclusions and Perspectives

The role of carriers in drug import was, until recently, overlooked in favor of the idea of drug uptake by diffusion via cell membrane, even though extensive evidence to the contrary has been shown [3,12,53]. Despite the majority of studies regarding cellular drug uptake via membrane carriers has been performed based on human cells, several studies also suggest that transport through protein carriers rather than simple diffusion is likely to be the main route of cellular entry for many drugs in different human pathogens [16,17,20]. Nevertheless, knowledge of which transporters import which drugs remain scarce.

Antifungal agents used to treat systemic fungal infections can be grouped into four classes based on their site of action in pathogenic fungi: polyenes, azoles, echinocandins, and nucleoside analogs [99]. Azoles and the polyene amphotericin B target the fungal cell membrane, azoles inhibiting the ergosterol biosynthesis enzyme Erg11 while amphotericin B interacting directly with ergosterol molecules at the plasma membrane. Echinocandins, on the other hand, target the fungal cell wall, through the inhibition of β-1,3-glucan synthase, while the pyrimidine analogue 5-FC targets DNA, RNA, and protein synthesis [100]. A wide variety of topical agents belonging to different classes of antifungals are available as creams, ointments, gels, lotions, powders, shampoos, and other formulations [99,101]. In clinically relevant fungal pathogens, carrier-mediated transport has been associated to the well-known drugs, such as the azoles (unknown transporter(s)) and flucytosine (*Candida* and *Cryptococcus* Fcy2, *Aspergillus* FcyB), and also to the novel antifungals VL-2397 (*Aspergillus fumigatus* Sit1), BQM (*Candida albicans* Mdr1). Although azoles were demonstrated to enter fungal cells through membrane protein carriers, identification of those transporters in fungal pathogens was not accomplished so far. Nevertheless, hints can be retrieved from *Saccharomyces cerevisiae*, in which both the myo-inositol transporter, Itr1, and the purine-cytosine permease, Fcy2, were associated with azole uptake.

Mansfield et al. [16] gathered evidence suggesting that both imidazoles (ketoconazole) and triazoles (fluconazole, itraconazole, posaconazole, and voriconazole) are uptaken into *Saccharomyces cerevisiae*, *Candida albicans*, *Candida. krusei*, and *Cryptococcus neoformans* cells through carrier-mediated transport. However, these authors were not able to find a significant candidate, which might be due to the presence of multiple transporters equally capable of importing fluconazole, the presence of paralog genes, and transporter belonging to a gene family; in these cases, only multiple mutants would show resistance to the drugs; transporter being an essential gene and, therefore, is not present in single deletion collection. Azole drug uptake have been suggested in *Candida lusitaniae* as well, in a competitive fashion with 5-FC, but no transporter could be assigned to this observation [102].

In order to overcome this issue, it is worth inspecting what is known in other species regarding drug import. Lanthaler and co-authors [5] identified different transporters responsible for the uptake of fluconazole, ketoconazole, and clotrimazole in *Saccharomyces cerevisiae*, as well as other transporters that have an indirect effect on these drugs’ efficacy. For instance, robot-assisted experiments indicated that the myo-inositol transporter Itr1 mediates the import of all three azoles, along with the fatty-acid transporter Fat1 for ketoconazole and with the purine-cytosine transporter Fcy2 for fluconazole. Additionally, the iron transporters Ftr1 and Fet3 were found to have an indirect effect on both fluconazole and clotrimazole efficacy. Azole’s target (cytochrome P450 enzyme 14-α-demethylase) is a heme-containing protein; therefore, their effect could be due to some influence on the drug’s target. Nevertheless, effect on ketoconazole resistance was not observed. These clues obtained from the model yeast might leverage the search for azoles transporter(s) in pathogenic fungi.

Similarly to azoles, the polyene amphotericin B also targets the fungal cell membrane. Instead of impairing normal ergosterol biosynthesis, this drug binds to ergosterol in the membrane promoting cell permeabilization [103]. Nonetheless, amphotericin B can also induce the accumulation of ROS, resulting in DNA, protein, mitochondrial, and membrane damage [104], which means that it also acts intracellularly. The process through which amphotericin B enters fungal cells is unknown.

Echinocandins are the only new class of antifungals to reach the clinic in decades, with three echinocandins currently available for clinical use [105]. These antifungals are the first class of clinically useful antifungal drugs that target the fungal cell-wall. They act by acting as non-competitive inhibitors of β-(1,3)-d-glucan synthase enzyme complex which catalyses the production of glucan, the major component fungal cell walls, disrupting cell wall synthesis [106]. Given that this class of drugs has an intracellular target, they need to enter fungal cells to exert their function. The way it happens is not known, although there is evidence reinforcing the idea that they might be imported by membrane carriers [54], as it happens for azoles. Since echinocandins are the recommended first-line treatment against systemic candidiasis nowadays and resistant isolates are increasing, understanding the full mode of action of these drugs could provide a ground start to delineate new strategies to fight these numbers.

It is important to highlight that the role of specific carriers as drug importers has been suggested, mostly, based on decreased susceptibility registered upon the deletion of the encoding gene (Figure 12). Although it provides valuable clues, this is not the most direct approach to infer a role in drug uptake. Indeed, the deletion of a transporter encoding gene may impair drug accumulation through indirect processes, linked to the physiological role of these predicted drug importers, as is probably the case of *Saccharomyces cerevisiae* strains lacking Frt1 or Fet3 [5]. Drug transport measurements are the most accurate way to determine the role of a membrane transporter as a specific drug importer. For instance, the role of Mdr1 as BQM importer was determined by relative BQM accumulation, measured by its fluorescence [38]. Nevertheless, that kind of studies remain scarce.

Despite the contribution of passive diffusion to drug entry into fungal cells, reduced or modified drug import due to mutations in drug uptake transporters may explain why some pathogenic fungi are more resistant than others, namely in the increasing number of drug-resistant isolates in which the resistance mechanism is unknown. In fact, altered drug uptake has been shown to be a mechanism of resistance in different species of human pathogens [16,97]. Hence, in the interest of public health, there is an urgent need to better characterize the exact mechanism of action of a drug, including how it enters pathogenic cells. This is a challenging but essential task, not only to try to reduce the incidence of drug-resistant clinical isolates, but also to give insight into drug discovery and design.

## Figures and Tables

**Figure 1 genes-11-01324-f001:**
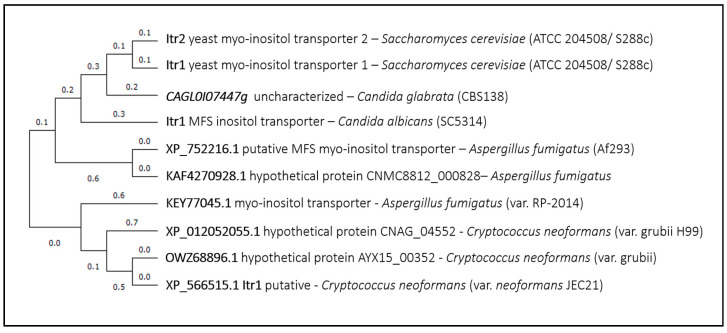
Itr1 phylogenetic analysis. The tree with the highest log likelihood is shown. This analysis involved 10 amino acid sequences. There were a total of 635 positions in the final dataset. Evolutionary analyses were conducted in MEGA X (https://pubmed.ncbi.nlm.nih.gov/29722887/).

**Figure 2 genes-11-01324-f002:**
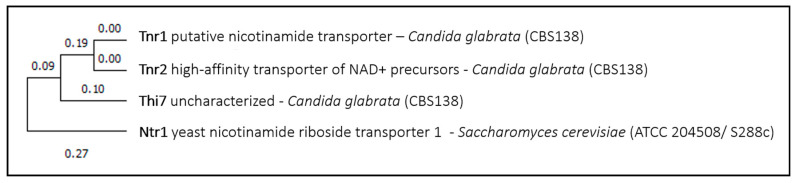
Nrt1 phylogenetic analysis. The tree with the highest log likelihood is shown. This analysis involved four amino acid sequences. There were a total of 596 positions in the final dataset. Evolutionary analyses were conducted in MEGA X.

**Figure 3 genes-11-01324-f003:**
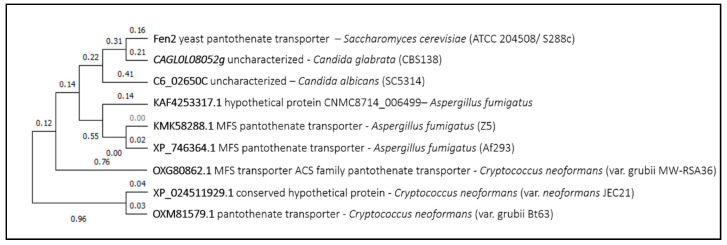
Fen2 phylogenetic analysis. The tree with the highest log likelihood is shown. This analysis involved nine amino acid sequences. There were a total of 650 positions in the final dataset. Evolutionary analyses were conducted in MEGA X.

**Figure 4 genes-11-01324-f004:**
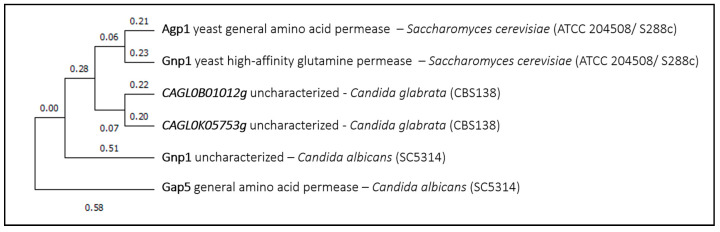
Agp1 phylogenetic analysis. The tree with the highest log likelihood is shown. This analysis involved six amino acid sequences. There were a total of 680 positions in the final dataset. Evolutionary analyses were conducted in MEGA X.

**Figure 5 genes-11-01324-f005:**
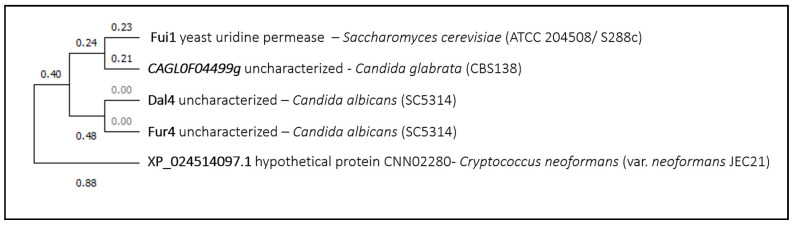
Fui1 phylogenetic analysis. The tree with the highest log likelihood is shown. This analysis involved five amino acid sequences. There were a total of 675 positions in the final dataset. Evolutionary analyses were conducted in MEGA X.

**Figure 6 genes-11-01324-f006:**
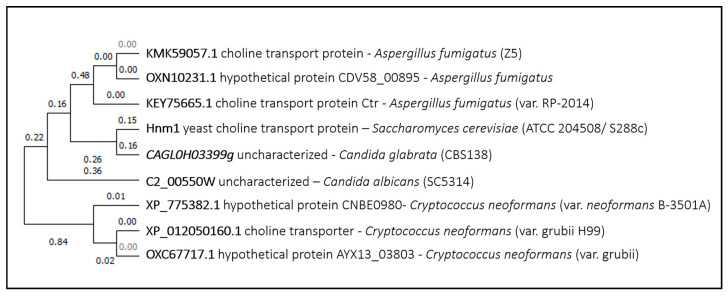
Hnm1 phylogenetic analysis. The tree with the highest log likelihood is shown. This analysis involved nine amino acid sequences. There were a total of 686 positions in the final dataset. Evolutionary analyses were conducted in MEGA X.

**Figure 7 genes-11-01324-f007:**
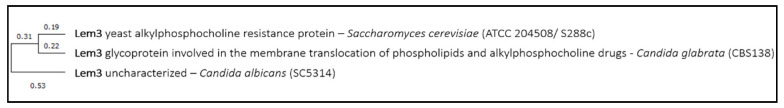
Lem3 phylogenetic analysis. The tree with the highest log likelihood is shown. This analysis involved three amino acid sequences. There were a total of 442 positions in the final dataset. Evolutionary analyses were conducted in MEGA X.

**Figure 8 genes-11-01324-f008:**
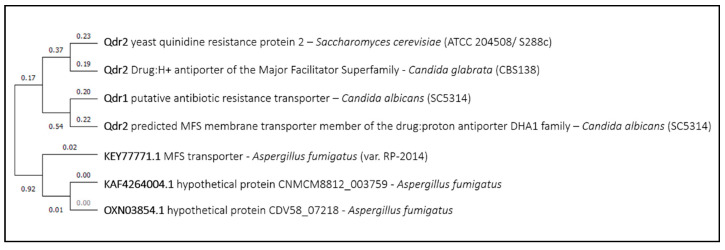
Qdr2 phylogenetic analysis. The tree with the highest log likelihood is shown. This analysis involved seven amino acid sequences. There were a total of 941 positions in the final dataset. Evolutionary analyses were conducted in MEGA X.

**Figure 9 genes-11-01324-f009:**
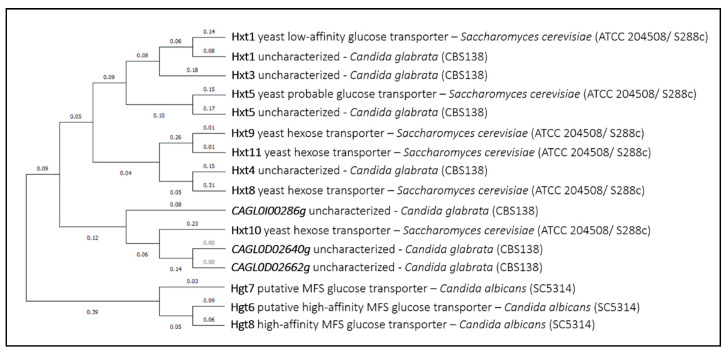
HXT (Hexose transporters) phylogenetic analysis. The tree with the highest log likelihood is shown. This analysis involved 16 amino acid sequences. There were a total of 594 positions in the final dataset. Evolutionary analyses were conducted in MEGA X.

**Figure 10 genes-11-01324-f010:**
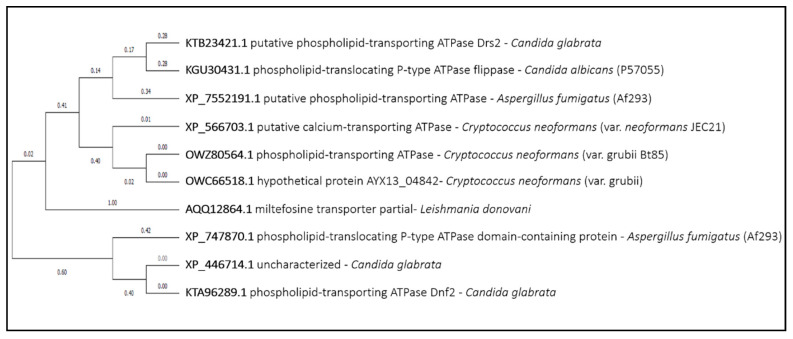
Miltefosine transporter phylogenetic analysis. The tree with the highest log likelihood is shown. This analysis involved 10 amino acid sequences. There were a total of 1645 positions in the final dataset. Evolutionary analyses were conducted in MEGA X.

**Figure 11 genes-11-01324-f011:**
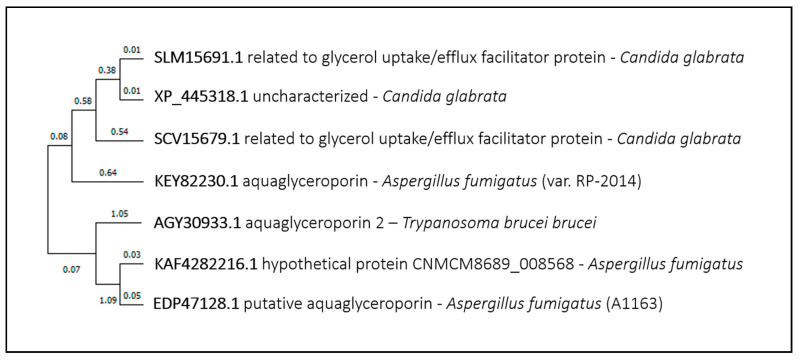
TbAqp2 phylogenetic analysis. The tree with the highest log likelihood is shown. This analysis involved seven amino acid sequences. There were a total of 684 positions in the final dataset. Evolutionary analyses were conducted in MEGA X.

**Figure 12 genes-11-01324-f012:**
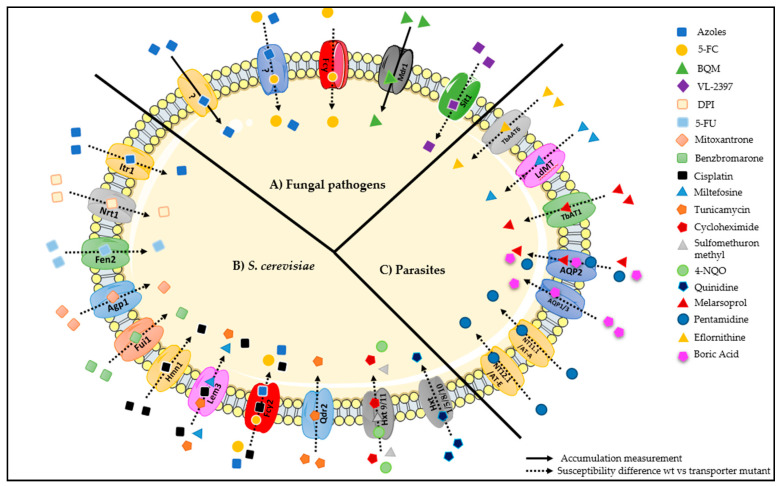
Drug importers in *Saccharomyces cerevisiae* and in pathogenic fungi (*Candida*, *Aspergillus* and *Cryptococcus*), and parasites (*Leishmania* and *Trypanosoma*). (**A**) Azole drugs enter into fungal cells by carrier-mediated transport in *Candida albicans*, *Candida. krusei*, *Cryptococcus neoformans*, and *Aspergillus fumigatus*, although the responsible transporter(s) is unknown [16]. In *Candida lusitaniae*, azole drug uptake in a competitive fashion with 5-FC has been suggested, but no transporter could be assigned to this observation [102]. The purine/cytosine permease Fcy2/FcyB is responsible for 5-FC uptake in *Candida*, *Cryptococcus,* and *Aspergillus* [21,23,29,31,32,33,34,35]. In *Candida albicans*, Mdr1 was demonstrated to be responsible for the uptake of BQM [38]. In *Aspergillus fumigatus*, VL-2397 uptake was shown to be catalyzed by an importer for ferrichrome-type siderophores, Sit1. (**B**) *Saccharomyces cerevisiae* importers of antifungal molecules or molecules that have showed antifungal activity, described in Section 3.1. (**C**) *Leishmania* and *Trypanosoma* importers of antifungal molecules or molecules that have showed antifungal activity, described in Section 3.2.

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
