# Peer review of "Carrier-Mediated Drug Uptake in Fungal Pathogens"

_genes, 2020, doi:10.3390/genes11111324_

Round 1
Reviewer 1 Report
Authors wrote very interesting review, which brings new information about uptake of antifungal drugs in yeast cells. This review summarized not only up-to-date knowledge about transporters, which play role in uptake of several antifungal drugs, but also based on their own phylogenetic analyses highlighted possible involvement of several other transporters in this phenomenon. The review fills empty niche in this scientific area and will be highly beneficial to scientific community, because studies about drug uptake are very rare. It is a widely accepted idea about dominant mechanism of entry of drugs into fungal cells via passive diffusion through the plasma membrane and there are only very few documented examples of drugs, which enters into yeast cells due to action of specific transporters, i.e. 5-flucytosine is transported by a cytosine permease. The article has high scientific quality but I suggest several minor (predominantly formal) changes, which should be taken into consideration.
Minor comments:
Lines 275 – 277: Susceptibility should be replaced by resistance according to the meaning of the sentence.
The main text is complemented by 11 schemes of phylogenetic analyses and one summarizing figure. In contrast to the summarizing figure 12, phylogenetic analyses (figures 1 – 11) are not so well-arranged and important information are a little bit lost. I suggest some improvement of layout, e.g. use of different types of writing (names of organisms in italics), highlight names of proteins (bold or different colour, but according to the convention should be written in small letters with the first capital letter) and distinguish main information from the rest descriptive information. Include all necessary information in the same order in all compared cases for higher clarity. Why are some information duplicated, e.g. the second and third sequences in figure 3 (similarly fig. 4, 5, 6, 9)? Why are not the names of organisms mentioned in all compared sequences? In figure 6, there is a typo in cerevisiae. In figure 10, the numbers overlap in the tree.
The legends of these figures are obviously copied, therefore in figure 8 are mentioned 3 amino acid sequences (copied from figure 7) instead of 7 sequences. The number of total positions in the final dataset – 442 is the same in both figures, which seems to me improbable.
Line 326. use Greek letter β
Line 381: BLASTP, in contrast to BLASTp mentioned at lines 141 – 148. If the meaning is identical, description should be uniform.
Line 471: itraconazole
Lines 482 – 483: double dots
List of references need careful check, e.g. all years of publication should be written in bold (references 4, 6, 7, 20, 44, 45, 53, 70, 81, 82, 85, 90, 92, 93, 98), names of species and genes or in vivo/in vitro should be written in italics (e.g. references 25, 32 – 34, 48, 49, 53, 56, 70, 73, 75 – 77, 83, 85, 86, 88), superscript (H+) – references 62, 64 and 65. There are some mistakes in names of authors (ref. 4, 78) and missing necessary information in references 8 (it is a book chapter), 54, 57, 72 and 98.
Author Response
REVIEWER 1
Authors wrote very interesting review, which brings new information about uptake of antifungal drugs in yeast cells. This review summarized not only up-to-date knowledge about transporters, which play role in uptake of several antifungal drugs, but also based on their own phylogenetic analyses highlighted possible involvement of several other transporters in this phenomenon. The review fills empty niche in this scientific area and will be highly beneficial to scientific community, because studies about drug uptake are very rare. It is a widely accepted idea about dominant mechanism of entry of drugs into fungal cells via passive diffusion through the plasma membrane and there are only very few documented examples of drugs, which enters into yeast cells due to action of specific transporters, i.e. 5-flucytosine is transported by a cytosine permease. The article has high scientific quality but I suggest several minor (predominantly formal) changes, which should be taken into consideration.
Minor comments:
Lines 275 – 277: Susceptibility should be replaced by resistance according to the meaning of the sentence. - Corrected accordingly
The main text is complemented by 11 schemes of phylogenetic analyses and one summarizing figure. In contrast to the summarizing figure 12, phylogenetic analyses (figures 1 – 11) are not so well-arranged and important information are a little bit lost. I suggest some improvement of layout, e.g. use of different types of writing (names of organisms in italics), highlight names of proteins (bold or different colour, but according to the convention should be written in small letters with the first capital letter) and distinguish main information from the rest descriptive information. Include all necessary information in the same order in all compared cases for higher clarity. Why are some information duplicated, e.g. the second and third sequences in figure 3 (similarly fig. 4, 5, 6, 9)? Why are not the names of organisms mentioned in all compared sequences? In figure 6, there is a typo in cerevisiae. In figure 10, the numbers overlap in the tree. - Corrected accordingly
The legends of these figures are obviously copied, therefore in figure 8 are mentioned 3 amino acid sequences (copied from figure 7) instead of 7 sequences. The number of total positions in the final dataset – 442 is the same in both figures, which seems to me improbable. - Corrected accordingly
Line 326. use Greek letter β - Corrected accordingly
Line 381: BLASTP, in contrast to BLASTp mentioned at lines 141 – 148. If the meaning is identical, description should be uniform. - Corrected accordingly
Line 471: itraconazole - Corrected accordingly
Lines 482 – 483: double dots - Corrected accordingly
List of references need careful check, e.g. all years of publication should be written in bold (references 4, 6, 7, 20, 44, 45, 53, 70, 81, 82, 85, 90, 92, 93, 98), names of species and genes or in vivo/in vitro should be written in italics (e.g. references 25, 32 – 34, 48, 49, 53, 56, 70, 73, 75 – 77, 83, 85, 86, 88), superscript (H+) – references 62, 64 and 65. There are some mistakes in names of authors (ref. 4, 78) and missing necessary information in references 8 (it is a book chapter), 54, 57, 72 and 98. Corrected accordingly
Reviewer 2 Report
The presented review focuses on the issue that is rarely discussed yet important form the clinical perspective. It is well-written and comprehensive.
I would like to ask the Authors to write additional 2-3 sentences about evolution the fungi (e.g. what is the evolutionary distance between Candida, Aspergillus and Cryptococcus species and how they are related to each other).
Please check also the comments in the manuscript pdf file.

Author Response
The presented review focuses on the issue that is rarely discussed yet important form the clinical perspective. It is well-written and comprehensive.
I would like to ask the Authors to write additional 2-3 sentences about evolution the fungi (e.g. what is the evolutionary distance between Candida, Aspergillus and Cryptococcus species and how they are related to each other). - Corrected accordingly
Please check also the comments in the manuscript pdf file. - Corrected accordingly